# Screening of Differentially Expressed Microsporidia Genes from *Nosema ceranae* Infected Honey Bees by Suppression Subtractive Hybridization

**DOI:** 10.3390/insects11030199

**Published:** 2020-03-22

**Authors:** Zih-Ting Chang, Chong-Yu Ko, Ming-Ren Yen, Yue-Wen Chen, Yu-Shin Nai

**Affiliations:** 1Department of Biotechnology and Animal Science, National Ilan University, No. 1, Sec. 1, Shen Nung Road, Ilan 26047, Taiwan; 2Genomics Research Center, Academia Sinica, No. 128, Academia Road, Sec. 2, Nankang District, Taipei 115, Taiwan; 3Department of Entomology, National Chung Hsing University, No. 145, Xingda Road, Taichung 402, Taiwan

**Keywords:** Microsporidia, *Nosema ceranae*, honey bee, cDNA subtraction

## Abstract

The microsporidium *Nosema ceranae* is a high prevalent parasite of the European honey bee (*Apis mellifera*). This parasite is spreading across the world into its novel host. The developmental process, and some mechanisms of *N. ceranae*-infected honey bees, has been studied thoroughly; however, few studies have been carried out in the mechanism of gene expression in *N. ceranae* during the infection process. We therefore performed the suppressive subtractive hybridization (SSH) approach to investigate the candidate genes of *N. ceranae* during its infection process. All 96 clones of infected (forward) and non-infected (reverse) library were dipped onto the membrane for hybridization. A total of 112 differentially expressed sequence tags (ESTs) had been sequenced. For the host responses, 20% of ESTs (13 ESTs, 10 genes, and 1 non-coding RNA) from the forward library and 93.6% of ESTs (44 ESTs, 28 genes) from the reverse library were identified as differentially expressed genes (DEGs) of the hosts. A high percentage of DEGs involved in catalytic activity and metabolic processes revealed that the host gene expression change after *N. ceranae* infection might lead to an unbalance of physiological mechanism. Among the ESTs from the forward library, 75.4% ESTs (49 ESTs belonged to 24 genes) were identified as *N. ceranae* genes. Out of 24 *N. ceranae* genes, nine DEGs were subject to real-time quantitative reverse transcription PCR (real-time qRT-PCR) for validation. The results indicated that these genes were highly expressed during *N. ceranae* infection. Among nine *N. ceranae* genes, one *N. ceranae* gene (AAJ76_1600052943) showed the highest expression level after infection. These identified differentially expressed genes from this SSH could provide information about the pathological effects of *N. ceranae*. Validation of nine up-regulated *N. ceranae* genes reveal high potential for the detection of early nosemosis in the field and provide insight for further applications.

## 1. Introduction

The honey bee (*Apis mellifera*) is an essential pollinator of many crop plants in natural ecosystems and agricultural crops; it contributes more than $14 billion to agriculture annually in North America [1,2,3,4]. However, the American apiculture industry experienced catastrophic losses of unknown origin since 2006. The decline and disappearance of the bee species in the natural environment and the collapse of honey bee colonies were defined as colony collapse disorder (CCD) [1]. The consequences of this syndrome are evident as an unexplained disappearance of adult bees, a lack of attention to the brood, reduced colony vigor, and heavy winter mortality without any apparent pathological infection. Reductions of honey bee populations were estimated at 23% over the winter of 2006–2007 [5] and at 36% over the winter of 2007–2008 [6]. Additionally, it leads to reduction of honey production and ultimately to colony collapse [7]. As a matter of fact, CCD is becoming a worldwide issue and honey bees are still suffering threats from either the natural environment (i.e., pesticide residues) or diseases caused by microorganisms (i.e., viruses, bacteria, and fungi). 

As aforementioned, honey bee pathogens, such as viruses, microsporidian, mites, and parasites, undoubtedly affect the development of honey bee colonies and are involved in CCD. Among these pathogens, microsporidia is one of the most common prevalences in bee populations [5,8]. There are two species of honey bee microsporidia: *Nosema apis* and *N. ceranae* [9]. Both are obligate intracellular parasitic fungi [10]. Microsporidia are the causation pathogens of nosemosis and usually cause chronic impacts on honey bee populations [9,11]. *N. apis* was first discovered in *A. mellifera* in Australia, North America, and Europe, and *N. ceranae* was originally found in *A. ceranae* in the 1990s [12,13]. From 2004 to 2006, *N. ceranae* infections were detected in *A. mellifera* in Taiwan and European countries [14,15]. Hereafter, the prevalence of *N. ceranae* was confirmed in the population of *A. mellifera*, and has become a globally distributed bee pathogen in this past decade [16]. *N. ceranae* coincided with early reports of CCD, and was suggested to be a factor in honey bee declines [5]. It affects adult bees and was recently found in collapsing *A. mellifera* colonies in Spain [17]. Experimental results suggest that *N. ceranae* is more virulent than *N. apis*; therefore, *N. ceranae* is considered to be an important bee pathogen that causes bee colony loss [16].

*N. ceranae* is a gut-pathogen, in terms that the *N. ceranae* infection typically causes cytopathic effects (CPE) in the infected midgut tissue of honey bees [11]. It was reported that an *N. ceranae* infection led to the reduction of honey production, malnutrition, shorter life span, and higher mortality of adult honey bees [7,11,18,19,20]. Moreover, the *N. ceranae* infection would change the carbohydrate metabolism and suppress metabolism related gene expression [21,22]. However, it is not easy for beekeepers to note *N. ceranae* infection in honey bee colonies due to the lack of obvious symptoms from latent infection of *N. ceranae*. Ignorance of latent infection of *N. ceranae* might raise the risk of long-term colony infection of *N. ceranae*, and then cause colony disease outbreaks [16]. Therefore, early detection of nosemosis at the latent infection stage is needed for management of honey bee colonies. 

In this study, we attempt to identify the differential expressed genes between *N. ceranae* infection and non-infected honey bees, and to clarify the gene expression profile of highly expressed genes during an *N. ceranae* infection. To this aim, a method of suppression subtractive hybridization (SSH), which is a powerful method for selectively amplifying differentially expressed genes [23], was used for identification of the differential expression *N. ceranae* genes, as well as the host response genes. The highly expressed *N. ceranae* specific genes were further validated and investigated. These data could provide insight for the detection of early nosemosis in the field.

## 2. Materials and Methods

### 2.1. Purification of Microsporidia from Infected Honey Bee Midguts

Spores for the inoculation were isolated from live heavily infected bees of *Apis mellifera*, collected from a naturally infected hive located in the experimental apiary at the National Taiwan University campus in Taipei, Taiwan, and the National Ilan University in Yilan, Taiwan. After dissecting the intestinal tracts, the midgut tissues were macerated in distilled water using a manual tissue grinder and examined under phase-contrast microscopy (IX71, Olympus, Tokyo, Japan). Infected tissues were homogenized and filtered through three layers of stainless net and centrifuged at 3000× *g* for 15 min. The pellet was then transferred into aseptic 90% Percoll and 10% 1.5 M NaCl, followed by centrifugation at 15,000× *g* for 40 min. The band centrifuged to the bottom of the tube contained mature spores [24,25]. The purified spores were stored in distilled water at 4 °C. The spore concentration was determined by counting with a hemocytometer chamber and the suspension was freshly prepared before using.

### 2.2. Experimental Infection

The experimental infection was modified from Higes et al. (2007), briefly described below; frames of sealed brood were obtained from a healthy colony of *A. mellifera* (Nosema-free, confirmed by PCR, weakly), and were kept in an incubator at 32 ± 2 °C and 95% relative humidity to provide newly emerged Nosema-free honey bees. Three days after eclosion, the bees were starved for 2 h and 100 bees per group were each fed with 5 μL of 50% sucrose solution containing 1.25 × 10^5^ spores of *N. ceranae,* by a droplet, with the spore solution at the tip of a micropipette, until it had consumed the entire droplet [26]. Control honey bees were fed with 5 μL of 50% sucrose solution. The honey bees were reared in an incubator at 32 ± 2 °C and 95% relative humidity, and the survival individuals were recorded daily for 21 days. The Kaplan–Meier survival curve was generated by SPSS software. For the sample collections, 10 bees from each cage were collected at 7, 14, and 21 days post infection (dpi) and their ventriculi processed for microscopic examination. Spores were counted and spore production was recorded every week until 3 weeks pi. Control bees were also sacrificed at days 7, 14, and 21 dpi, and analyzed to confirm the absence of spores in ventriculus.

### 2.3. Detection of Infection

DNA was extracted from the midgut tissue of control bees and infected honey bees by using the following procedure modified from Tsai et al., 2002 [27]. In brief, midgut tissues from 10 control and infected honey bees were homogenized in TE buffer (0.1 M Tris, 0.01 M EDTA, pH 9.0) with a pestle. The macerated solution was centrifuged and the supernatant was discarded. The pellets were incubated with RNase A at 37 °C for 1 h and proteinase K at 56 °C overnight, followed by DNA extraction with an equal volume of phenol-chloroform-isoamyl alcohol (25:24:1), and DNA precipitation with 1/10 volume of 3 M sodium acetate, and 2 times the volume of pure ethanol. The DNA pellet was air dried and then dissolved in water. The small subunit ribosomal DNA (SSU rDNA) fragment of the microsporidium was amplified using primer set 18f /1537r (Appendix A). For the internal control, 18S primer set 143F/145R was used (Appendix A). Each 50-μL PCR mix contained 5 μL 10× reaction buffer (Bioman), 4 μL 2.5 mM dNTPs, 0.5 μL 100 mM of each primer, 1 μL 1.25 U HiFi *Taq* polymerase (RBC), and 1 μL template DNA. PCR amplifications were performed as follows on an Primus 96 Plus Thermal Cycler (MWG-Biotech, Ebersberg, Germany): thermal cycler was preheated at 95 °C for 5 min, 35 cycles at 94 °C for 30 s, 50 °C for 1 min, and 72 °C for 2 min, followed by a 10 min final extension at 72 °C, and storage at 20 °C. The PCR product was cloned into T&A cloning vector (RBC Bioscience) and commercially sequenced (Genomics Biosci. & Tech. Company, New Taipei, Taiwan).

### 2.4. Construction of Subtractive Complementary DNA (cDNA) Library

Total RNA was extracted using TRIzol^®^ Reagent (ambion), according to the manufacturer’s instructions, from 10 of the control bee and infected bee tissues, at 7, 14 and 21 dpi, and then equal quantities of three RNA samples, per group, were pooled together to maximize detection of differential expression of genes that may show variation between individuals in the exact timing of expression. Single-stranded and double-stranded cDNA were synthesized from the infected and control RNA pools with the SMART^TM^ PCR cDNA Synthesis Kit (Takara, Kyoto, Japan), according to the manufacturer’s protocol.

Two subtractive cDNA libraries (forward and reverse subtractive cDNA libraries) from the control and infected honey bees, were constructed by using the PCR-Select cDNA Subtraction kit (Clontech), according to the manufacturer’s specifications. Briefly described below, cDNA from the infected honey bees was used as the tester (forward subtractive cDNA library, fscl)/driver (reverse subtractive cDNA library, rscl), and cDNA from the control honey bees as the driver (fscl)/tester (rscl). Both tester and driver cDNA were digested with *Rsa*I to produce shorter, blunt-ended fragments. After digestion with *Rsa*I, the tester cDNA was divided into two portions, each of which was ligated with a different adapter (Adaptor-1 and Adaptor-2R) at 16 °C for overnight. After ligation, each tester cDNA was separately hybridized at 68 °C for 8 h with an excess of driver cDNA after denaturation at 98 °C for 90 s. Then, the two hybridized samples were mixed together and hybridized at 68 °C overnight, with excess of denatured driver cDNA. The resulting mixture was added with 200 mL dilution buffer and amplified by two rounds of suppression PCR. Before the primary PCR, the reaction mixture was incubated at 75 °C for 5 min to extend the adaptors. Primary PCR was performed at 94 °C for 30 s, 66 °C for 30 s, and 72 °C for 90 s for 27 cycles, in a reaction volume of 25 μL. The PCR product was then diluted 10-fold, and 1 μL-diluted product was used as template in the next subsequently nested PCR. The subtracted secondary PCR products were purified with DNA Clean/Extraction Kit (GeneMark) and then ligated into T&A cloning vector (RBC) followed by transformation into *Escherichia coli*, DH5α competent cells (RBC), to generate subtracted cDNA libraries. Each step of SSH was tested following the manufacturer’s instructions. A total of 192 white colonies from forward (96 colonies) and reverse libraries (96 colonies) were checked by colony PCR. The PCR products (insert DNA > 100 bp) were further subject to the dot-blotting method, which were described from the manufacturer’s protocol, and the potential colonies were identified and commercially sequenced (Genomics Bioscience & Technology, New Taipei, Taiwan). DNA sequencing from the 3′ end and 5′ end of the cDNA was conducted with M13 forward or reverse primers, respectively, on a high throughput automated sequencer (MJ Research BaseStation and ABI3730) (Thermo Fisher Scientific Inc., Waltham, MA, USA), using standard protocols.

### 2.5. DNA Sequences Analysis

cDNA sequences from each SSH library was sequenced. CodonCode Aligner was used for base-calling (Q > 13), trimming vector sequences, and sequence assemblage. All of the sequences were filtered for size (less than 100 bp). Putative functions of the unique sequences were discovered by using BLASTn and BLASTp to translate each nucleotide query sequence into all reading frames and then search for matches in the National Center for Biotechnology Information (NCBI) non-redundant (nr) database. Significant hits (with *E* value < 10^−10^) in the NCBI nr database were further submitted to the Gene Ontology (GO) enrichment analysis (http://geneontology.org/page/go-enrichment-analysis) for protein functions. The Gene GO annotations classify proteins by molecular function, biological process, and cellular component. Each unique sequence was tentatively assigned GO classification based on annotation of the E-value for the “best hit” match. These data were then used to classify the corresponding genes, according to their GO functions.

### 2.6. RT-qPCR Validations of Microsporidia Specific Genes

To further validate the differential expressed microsporidia specific genes in *N. ceranae* infected honey bees, microsporidia genes have 2 ≥ expressed sequence tags (ESTs) present in the forward library, and dot-blotting difference ≥ 3-fold were selected and subject to quantitative RT-qPCR analysis. Gene specific primer sets for RT-qPCR were designed by Primer Express v3.0 (Appendix A). Total RNA was extracted using TRIzol^®^ Reagent (ambion), according to the manufacturer’s instructions, from 5 of the control and infected bee midguts at 7, 14, and 21 dpi. The extracted RNA (2 μg) was treated with DNaseI (Roche Molecular Biochemicals, Basel, Switzerland), recovered by a phenol/chloroform/isoamyl alcohol extraction, and precipitated with ethanol. The RNA was then treated with DNase I (Invitrogen Life Technologies, Waltham, MA, USA), following the manufacturer’s instructions, to reduce genomic DNA contamination before RT-qPCR. The DNase I-treated total RNA samples were conducted to the reverse-transcription by GScript RTase kit (GeneDireX, New Taipei, Taiwan), following the manufacturer’s instructions. The reaction mixtures were incubated at 42 °C for 1.5 h and then the reaction was stopped at 70 °C for 15 min. Real-time qPCR was performed by using Thermo Scientific Verso SYBR Green 1-step qRT-PCR ROX Mix kit (Thermo Fisher Scientific Inc., Waltham, MA, USA) in a 96-well Bio-Rad CFX96 Real-Time PCR System (Bio-Rad, Hercules, CA, USA). The qRT-PCR was performed as follows: 95 °C for 3 min, 40 cycle at 95 °C for 10 s, 59 °C for 30 s, 95 °C for 10 s. All samples were performed in triplicate. The relative gene expression levels were calculated by 2^−△△Ct^ method [28]. 

## 3. Results and Discussion

### 3.1. Microsporidian Infection

The midgut tissues of the control and infected honey bees were homogenized and mature spores in the macerated solution were counted under microscopy. The formation of mature spores was observed after 14 to 21 dpi; moreover, the spore number increased from 14 to 21 dpi (Figure 1A,B). The DNA of honey bee midgets was extracted for confirming the infection of *N. ceranae* by PCR with 18f and 1537r primer sets (microsporidia degenerated primer set). The results of PCR revealed that only the infected honey bees were detected (Figure 1D). The infection of *N. ceranae* was detected at 7 dpi, though the mature spores were only observed at 14 dpi (Figure 1). This result was similar to the previous report that different parasite stages of *N. ceranae* should present in the midgut epithelial cells after the experimental infection of *N. ceranae* with honey bees at 6–7 dpi [11], indicated the honey bees were under *N.*
*ceranae* infection. It has been described that the pathological effects of *N. ceranae* infected *A. mellifera* resulted in the reduction of the lifespan [11,21]. From our survival rate analysis, adult honey bees infected with *N. ceranae* showed lower survival possibility and survival time than those uninfected honey bees (Figure 1C). These data proved a consistent negative effect of the *N. ceranae* to adult bees. 

### 3.2. Screening of the Subtracted cDNA libraries

Two subtracted cDNA libraries were established, and a total of 192 white colonies (96 colonies each) were first screened by colony PCR (Figure 2). The results of colony PCR showed that the positive rate was 97.9% for fscl and 95.8% for rscl. The size of inserted cDNA fragments varied from 180 bp to 1 kb (Figure 2). The positive PCR products were then subjected to dot blotting for screening the differential ESTs. A total of 112 differentially expressed sequence tags (ESTs) were identified in the subtracted libraries and sequenced. Moreover, 65 and 47 ESTs were identified from infected (forward) and non-infected (reverse) subtracted libraries, respectively (Table 1 and Figure 2B,C,E,F). These clones revealed the genes with high differential expression levels and were further sequenced.

### 3.3. Analysis of EST Sequences

A total of 112 ESTs were sequenced and further analyzed. The average sequence length of ESTs in each library was 426 bp for the infected (forward) library and 395 bp for the non-infected (reverse) library (Table 1). According to the NCBI BLASTn analysis, 75.4% ESTs (49 ESTs) were identified as microsporidia genes and 20% ESTs (13 ESTs) belonged to *Apis* spp. in the fscl (Table 2 and Figure 2B,C). From the rscl, 93.6% ESTs (44 ESTs) showed significant homology to the insect host, particularly Hymenoptera insects (*A. mellifera* and *A. florea*) (Table 3 and Figure 2E,F). This result indicated that the microsporidia genes were highly expressed during the infection process. Only a few host genes showed dramatic responses of up-regulated during *N. ceranae* infection, while most of the up-regulated host genes were identified from the rscl, suggesting the suppression of host gene expression by *N. ceranae* infection. Based on this result, several immune-related genes (i.e., *Jra*, *pi3k*) of honey bees were identified from the rscl. As a matter of fact, that *N. ceranae* infection could change the metabolism of the host and lead to negative effects on the immune systems of honey bees [29]. It has also been reported that honey bees infected with *N. ceranae* showed a down-regulation of some immune-related genes, such as *abaecin*, *apidecin*, *defensin*, *hymenoptaecin*, *glucose dehydrogenase* (*GLD*), and *vitellogenin* (*Vg*), suggesting that *N. ceranae* infection suppresses immune defense mechanisms in honey bees [30,31,32].

The result of Gene Ontology (GO) analysis showed a total of 30 differentially expressed genes (DEGs) (78.9%) from the host were identified and further categorized into molecular function, biological process, and cellular component (Figure 3; Appendix A). Moreover, 16 GO terms belonged to three major categories (Figure 3; Appendix A). Based on the GO analysis, the genes involved in catalytic activity (GO:0003824) were highly up-regulated and were followed by cellular process (GO:0009987)/membrane (GO:0016020), and binding (GO:0005488)/transporter activity (GO:0005215)/cell (GO:0005623) in the fscl. For the rscl, host genes involved in the metabolic process (GO:0008152) and cell (GO:0005623) showed high changes during *N. ceranae* infection, followed by catalytic activity (GO:0003824)/binding (GO:0005488)/cellular process (GO:0009987)/membrane (GO:0016020), and transporter activity (GO:0005215)/organelle (GO:0043226) (Figure 3; Appendix A). In conclusion, according to our data, a high percentage of host DEGs were involved in catalytic activity (GO:0003824), cellular process (GO:0009987), membrane (GO:0016020)/cell (GO:0005623), and transporter activity (GO:0005215) (Figure 3; Appendix A). As a matter of fact, the up-regulation of the α-glucosidase gene, and genes that were involved in the trehalose transport, and the down-regulation of the trehalase and the glucose-methanol-choline oxidoreductase, three encoding genes during *N. ceranae* infection. The unbalance of gene expressions result in modifications of carbohydrate metabolism in infected honey bees; suggesting that these changes in gene expressions and carbohydrate metabolism were manipulative activity of the *N. ceranae* to get nutrients from the host [21,30]. Based on our data, the genes involved in the metabolic process (GO:0008152) from the rscl showed differential expression, revealed the host gene expression change after *N. ceranae* infection, and led to the unbalance of physiological mechanisms. 

### 3.4. Identification and Validation of Highly Expressed N. ceranae Specific Genes

From the forward (infected) library, 75.4% ESTs (49 ESTs belonged to 24 genes) were identified as *N. ceranae* genes (Table 2). Of 24 *N. ceranae* genes, nine DEGs, which have more than three identified ESTs and >3-fold change by dot bolt analysis, were selected for further validation by qRT-PCR, including four hypothetical proteins (AAJ76_1600052943, AAJ76_500092411, AAJ76_2000141845, and AAJ76_1900028347), *14-3-3 protein 1-like protein* (AAJ76_300017093), *Heat shock protein 90* (AAJ76_1170002375), *Ubiquitin* (AAJ76_1300017036), *Forkhead hnf3 transcription factor* (AAJ76_500091146), and *Histone H3* (AAJ76_800037613) (Figure 4). Similar to the expectations, the results indicated that these genes showed high expression during *N. ceranae* infection. Moreover, the results also showed that all microsporidia specific gene expressions had the highest gene expression level at 7 dpi, and decreased at 14 and 21 dpi (Figure 5). It should be noted that among these nine *N. ceranae* specific genes, a *hypothetical protein* (AAJ76_1600052943) (the previous library clone no. = sr22 and provisional named sr22 gene) has consistently the highest expression, which up-regulated to ~1.1 × 10^4^ fold at 7 dpi to the mock infection 2340, and 1148 fold at 14 and 21 dpi, respectively (Figure 5). For the other eight genes, three genes, *histone H3*, *ubiquitin*, and *hypothetical protein* (AAJ76_2000141845), showed higher expression levels, which ranged from ~2100 to 4400 fold to the mock infection at 7 dpi; the *hypothetical protein* (AAJ76_2000141845) also showed consistently high expression at 14 to 21 dpi (Figure 5). 

Ubiquitin is a small protein that is involved in the ubiquitin-proteasome system (UPS). The UPS is the crucial intracellular protein degradation system in eukaryotes. It has been reported that the infection of *N. bombycis* resulted in up-regulation of protein degradation-related enzymes in the host midgut, including the *S-phase kinase-associated protein 1,* and the ubiquitin-conjugating enzyme *E2G*. Both of which are involved in the ubiquitin-proteasome pathway [33]. From our data, the high expression of *ubiquitin* was observed in *N. cerana*, indicating the host defense mechanism to the parasite and, therefore, triggers the protein degradation-related enzyme in *N. cerana.*

Besides, the high expression of *histone H3* was detected in the *N. ceranae* infection process; histone proteins are essential for DNA packing, chromosome stabilization, and gene expression in the nucleus of a eukaryotes cell. Each nucleosome consists of two copies of each core histone—H2A, H2B, H3, and H4—to be an octameric protein complex. Histones H3 and H4 are the most highly conserved histones, and play a regulatory role in chromatin formation, such as retaining the ability to impede transcription [34]. Therefore, it was assumed that high expression of *N. ceranae histone H3* might enhance the regulation of the chromatin formation and facilitate the microsporidia DNA replication.

For the other up-regulated *N. ceranae* genes, they showed up-regulated expression profiles during *N. ceranae* infection (Figure 5). The functions of several genes revealed the response of *N. ceranae* to the honey bee host and also showed the replication activity of *N. ceranae* during infection. For the *14-3-3 protein 1-like protein* (AAJ76_300017093), it is a serine/threonine binding protein. The *14-3-3 protein 1-like protein* inhibits apoptosis through sequestration of pro-apoptotic client proteins [35]. Interestingly, the high expression levels of *ubiquitin* of *N. ceranae* was detected at 7 dpi and then decreased at 14 and 21 dpi, while the expression of *14-3-3 protein 1-like protein* was higher than that of *ubiquitin* at 21 dpi; besides, the *heat shock protein 90 (Hsp90*) was also identified as a high expressed *N. ceranae* gene that facilitates metastable protein maturation, stabilization of aggregation-prone proteins, quality control of misfolded proteins, and assists in keeping proteins in activation-competent conformations [36]; it seems that during the infection process, *N. ceranae* not only needs to against the stress from the host, but also attempts to stabilize the physical pathway itself for self-propagation purposes. This hypothesis is worthy to be addressed in the future.

The gene *forkhead hnf3 transcription factor* (AAJ76_500091146) presented throughout the animal kingdom, as well as in fungi and yeast (Mazet et al. 2003). The function of *forkhead hnf3 transcription factor* is characterized by a type of DNA-binding domain known as the forkhead box (FOX) [37]. It was also described that the *forkhead hnf3 transcription factor* is involved in a wide range of biological functions, including development, growth, stress resistance, apoptosis, cell cycle, immunity, metabolism, reproduction, and ageing [38]. As a matter of fact, in terms of these genes, it might play an important role for the replication of microsporidia, but it still needs to be further evaluated. The genome of the honey bee (*A. mellifera*) had been sequenced, and it consisted of a little more than 10,000 genes, lower than other insects (*D. melanogaster*, 13,600 genes; *A. gambiae*, 14,000 genes; *B. mori*, 18,500 genes) [39]. In additional, the draft assembly (7.86 MB), and highly compact (~1 gene/kb) genome of the *N. ceranae* genome, has been fully sequenced in 2009 [40]. *N. ceranae* has a strongly AT-biased genome (74% A+T) and a diversity of repetitive elements, complicating the assembly. A total of 2614 predicted protein-coding sequences were predicted and 1366 of these genes have homologs in the microsporidian *Encephalitozoon cuniculi*, the most closely related published genome sequence. These databases could be a reference for studying and clarifying the virulence of *N. ceranae* and the interaction between their host, *A. mellifera,* in the near future. 

In this study, the method of SSH was mainly used for the identification of the differential expression *N. ceranae* genes. From our data, several *N. ceranae* genes were identified as differentially expressed genes through the SSH, which could provide information about the pathological effects of *N. ceranae*. It should be also noted that the microRNA (miRNA) of *N. ceranae* might play important roles in the transcriptome of *N. ceranae* during infection, and reduction of the replication of *N. ceranae* via knockdown of transcripts was also demonstrated [41]. Thus, our data could provide the candidate genes for evaluation of nosemosis-controlling effects during infection. Since the condition (i.e., hybridization) of SSH might have some impacts on the result (i.e., the sequence through-put). Therefore, more modern methods, such as whole transcriptomics for deep sequencing, could be used for further investigations. In terms of the validation of nine up-regulated *N. ceranae* genes, the *N. ceranae* genes, which were identified by the SSH method, revealed high coordinates to the expectations. These genes showed high potential for the detection of early nosemosis in the field and provided insight for further applications.

## 4. Conclusions

In this study, 28 honey bee genes and 24 genes in *N. ceranae* were identified as DEGs after honey bees were infected with *N. ceranae* via the SSH approach. For the host DEGs, a high percentage of DEGs involved in catalytic activity and metabolic processes revealed that the host gene expression change after *N. ceranae* infection might lead to the unbalance of physiological mechanisms. Of 24 *N. ceranae* genes, nine DEGs were subject to real-time quantitative reverse transcription PCR (real-time qRT-PCR) for validation, and all of them showed high expression levels during different time post infections. Regarding the validation results, these genes revealed highly potential for the detection of different stages of nosemosis in the field. It should also be noted that one *N. ceranae* gene (AAJ76_1600052943, sr22) showed the highest expression level after infection. This gene might be further developed as a biomarker for early nosemosis detection. The data from this study could provide information on the pathological effects of *N. ceranae* and new insight for further applications on honey bee pathogen detections.

## 5. Patent

Taiwan, I481618. Lo, C. F., Wang, C. H., and Nai, Y. S. *Molecular Markers for nosema infection.*

## Figures and Tables

**Figure 1 insects-11-00199-f001:**
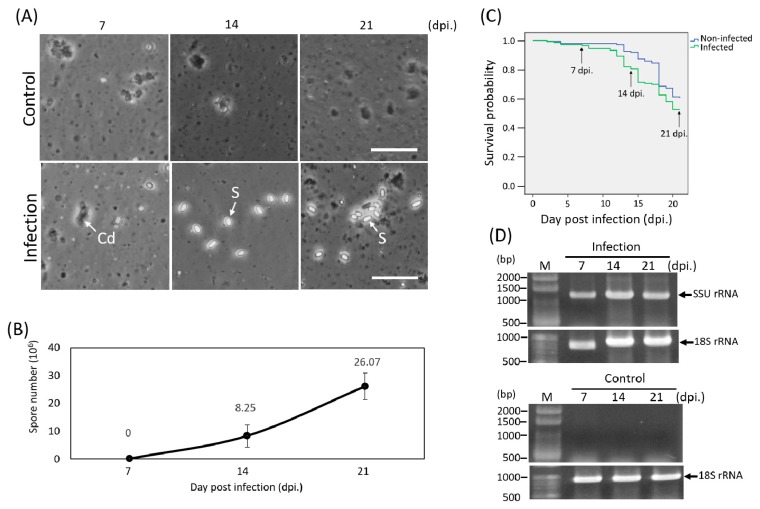
Observations and molecular detection of microsporidia infection at 7, 14, and 21 days post infection (dpi). (**A**) Light microscope observations; at 14 and 21 dpi, the mature spores (S) were formed. Cd = Cell debris; Scale bar = 25 μm; (**B**) spore number was counted at each time point; (**C**) Kaplan–Meier survival curve; (**D**) molecular detection of microsporidia infection; SSU rRNA = small subunit ribosomal RNA; 18S ribosomal DNA (rDNA) = DNA internal control. Control = non-infected honey infection = infected honey bee.

**Figure 2 insects-11-00199-f002:**
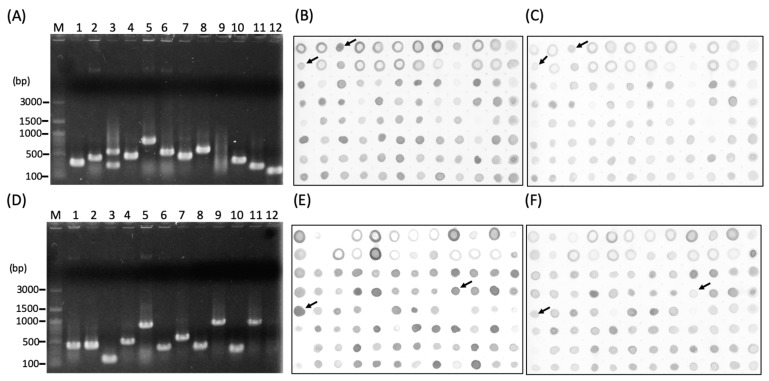
Screening and identification of differential expressed sequence tags (ESTs) from the *Nosema ceranae* infected honey bee by suppression subtractive hybridization (SSH) and Dot-blotting. ESTs insert size of (**A**) forward and (**D**) reverse SSH complementary DNA (cDNA) libraries were confirmed by colony PCR. (**B**) Forward subtractive library (fsl, differential ESTs in infected group) hybridized with forward-subtracted probe and (**C**) with reverse-subtracted probe. (**E**) Reverse subtractive library (rsl, differential ESTs in control group) hybridized with reverse-subtracted probe and (**F**) with forward-subtracted probe. M = 100 bp DNA ladder. The arrows indicated the differential ESTs for sequencing.

**Figure 3 insects-11-00199-f003:**
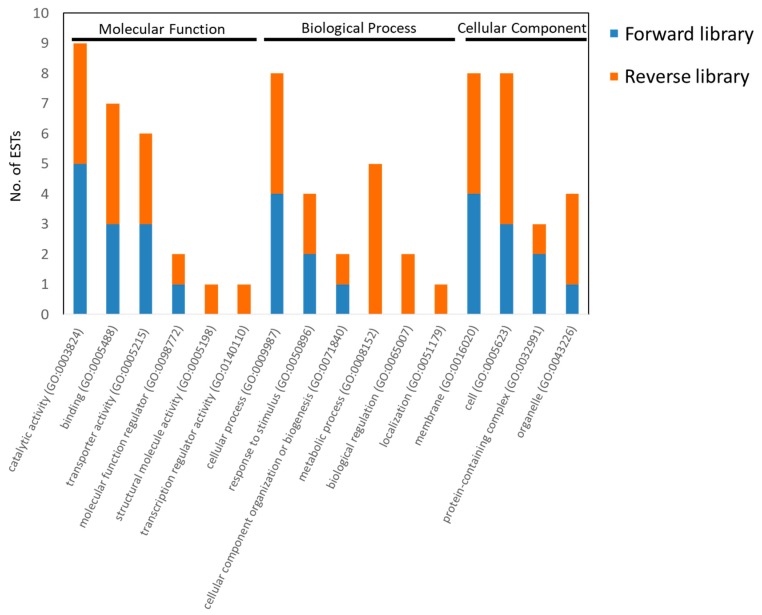
Gene Ontology (GO) analysis of 47 host insect ESTs from forward library (13 ESTs) and reverse library (34 ESTs).

**Figure 4 insects-11-00199-f004:**
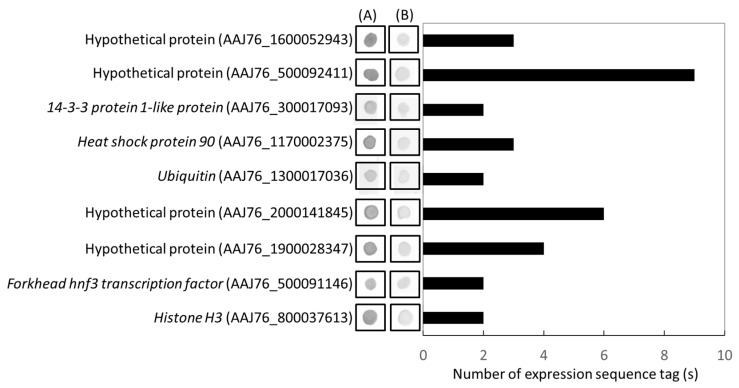
Distribution of expression sequence tags (ESTs) belong to eighteen *N. ceranae* specific genes and dot-blot hybridization test the cDNA library of *N. ceranae*-infected cDNA library. (**A**) The probes that came from subtractive PCR products of the *N. ceranae*-infected cDNA library was used to test the cDNA library of *N. ceranae*-infected cDNA library; (**B**) the probes that came from unsubtractive PCR products of the control honey bee cDNA library was used to test the cDNA library of the *N. ceranae*-infected cDNA library.

**Figure 5 insects-11-00199-f005:**
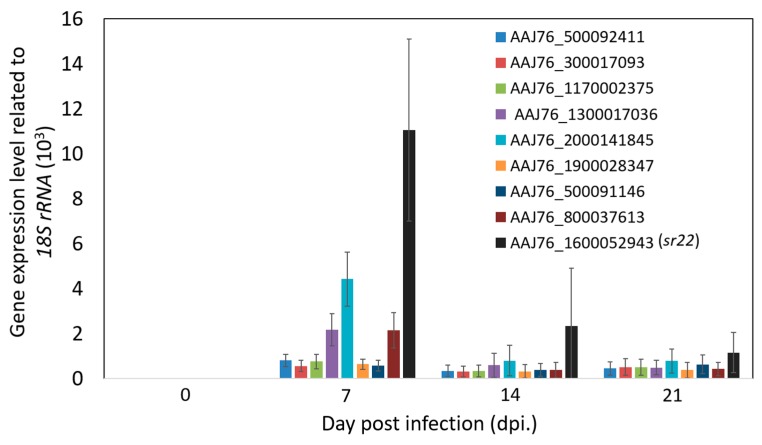
Validation of nine microsporidia specific gene expressions by RT-qPCR at 7, 14, and 21 dpi (the highest expressed *N. ceranae* gene, AAJ76_1600052943 (*sr22*).

**Table 1 insects-11-00199-t001:** Information for the expression sequence tags (ESTs).

Library Type	Value	Mean EST Sequence Length (bp)
Forward library	65	426
Reverse library	47	395
Total ESTs ^a^	112	411

^a^ ESTs = Expression sequence tags.

**Table 2 insects-11-00199-t002:** Analysis of 65 differential expressed sequence tags (ESTs) identified from forward cDNA library. NCBI = National Center for Biotechnology Information.

No	EST Number	Insert Sequence Length (bp)	NCBI_BLAST_N			NCBI_BLAST_X		
			Species	Gene Name	E-Value	Species	Gene Name	E-Value
1	SSH-Forward-lib-1	302	*Nosema ceranae*	Hypothetical protein (AAJ76_1600052943)	3.00E-127	*Nosema ceranae*	Hypothetical protein (AAJ76_1600052943)	9.00E-55
2	SSH-Forward-lib-2	396	*Nosema ceranae*	Dynein light chain 1 (AAJ76_5000122115)	4.00E-77	*Nosema ceranae*	Dynein light chain 1	2.00E-24
3	SSH-Forward-lib-3	302	*Nosema ceranae*	Hypothetical protein (AAJ76_1600052943)	2.00E-124	*Nosema ceranae*	Hypothetical protein (AAJ76_1600052943)	5.00E-55
4	SSH-Forward-lib-4	421	*Nosema ceranae*	Hypothetical protein (AAJ76_700047245)	1.00E-177	*Nosema ceranae*	Hypothetical protein NCER_101230 (AAJ76_700047245)	2.00E-64
5	SSH-Forward-lib-5	958	*Nosema ceranae*	60s ribosomal protein L10a (AAJ76_1200039265)	0.00E+00	*Nosema ceranae*	60s ribosomal protein L10a	2.00E-145
6	SSH-Forward-lib-7	413	*Nosema ceranae*	Hypothetical protein (AAJ76_500092411)	0.00E+00	*Nosema ceranae*	Actin	8.00E-86
7	SSH-Forward-lib-8	532	*Nosema ceranae*	Hypothetical protein (AAJ76_500092411)	0.00E+00	*Nosema ceranae*	Actin	1.00E-82
8	SSH-Forward-lib-9	346	*Nosema ceranae*	14-3-3 protein 1-like protein (AAJ76_300017093)	8.00E-158	*Nosema ceranae*	14-3-3 protein 1-like protein	4.00E-66
9	SSH-Forward-lib-10	354	*Nosema ceranae*	Heat shock protein 90 (AAJ76_1170002375)	2.00E-153	*Nosema ceranae*	Heat shock protein 90	7.00E-61
10	SSH-Forward-lib-11	262	*Apis mellifera*	PREDICTED: *Apis mellifera* protein G12-like (LOC102654405)	1.00E-100	*Apis mellifera*	PREDICTED: protein G12-like	1.00E-18
11	SSH-Forward-lib-12	181	*Apis mellifera*	*Apis mellifera* ribosomal protein LP1 (RpLP1) (NM_001185144)	8.00E-63	*Apis mellifera*	Unknown	6.00E-17
12	SSH-Forward-lib-13	326	*Nosema ceranae*	Ubiquitin (AAJ76_1300017036)	1.00E-115	*Nosema ceranae*	Ubiquitin	1.00E-47
13	SSH-Forward-lib-14	231	*Nosema ceranae*	Hypothetical protein (AAJ76_2000141845)	1.00E-88	*Nosema ceranae*	Hypothetical protein (AAJ76_2000141845)	8.00E-34
14	SSH-Forward-lib-15	212	*Nosema ceranae*	Hypothetical protein (AAJ76_2000141845)	3.00E-93	*Nosema ceranae*	Hypothetical protein (AAJ76_2000141845)	2.00E-33
15	SSH-Forward-lib-19	959	*Apis mellifera*	PREDICTED: *Apis mellifera* insulin receptor B (IR-B), transcript variant 1	0.00E+00	*Bacteroides dorei*	Hypothetical protein HMPREF1063_05178	1.00E-18
16	SSH-Forward-lib-20	163	*Nosema ceranae*	Hypothetical protein (AAJ76_500092411)	1.00E-47	*Nosema ceranae*	Actin	3.00E-16
17	SSH-Forward-lib-21	814	*Schistosoma rodhaini*	Schistosoma rodhaini genome assembly S_rodhaini_Burundi (LL960759 )	3.00E+00	-	-	-
18	SSH-Forward-lib-23	231	*Nosema ceranae*	Hypothetical protein (AAJ76_2000141845)	2.00E-93	*Nosema ceranae*	Hypothetical protein (AAJ76_2000141845)	6.00E-35
19	SSH-Forward-lib-25	319	*Nosema ceranae*	Mitochondrial sulfhydryl oxidase (AAJ76_600077605)	3.00E-136	*Nosema ceranae*	Mitochondrial sulfhydryl oxidase	9.00E-58
20	SSH-Forward-lib-26	133	*Alexandrium tamarense*	*Alexandrium tamarense* mRNA (AB233356)	4.00E-14	*Durinskia baltica*	Cytochrome oxidase subunit 3, partial	2.00E-32
21	SSH-Forward-lib-27	537	*Nosema ceranae*	Hypothetical protein (AAJ76_2000141845)	0.00E+00	*Nosema ceranae*	Hypothetical protein (AAJ76_2000141845)	2.00E-116
22	SSH-Forward-lib-29	186	*Apis florea*	PREDICTED: *Apis florea* V-type proton ATPase 16 kDa proteolipid subunit (LOC100867892)	5.00E-66	*Melipona quadrifasciata*	V-type proton ATPase 16 kDa proteolipid subunit, partial	2.00E-32
23	SSH-Forward-lib-30	244	*Nosema ceranae*	Hypothetical protein (AAJ76_6800017741)	4.00E-106	*Nosema ceranae*	Hypothetical protein (AAJ76_6800017741)	9.00E-40
24	SSH-Forward-lib-32	323	*Nosema ceranae*	Hypothetical protein (AAJ76_1600052943)	3.00E-135	*Nosema ceranae*	Hypothetical protein (AAJ76_1600052943)	1.00E-56
25	SSH-Forward-lib-34	885	*Apis mellifera*	PREDICTED: *Apis mellifera* early nodulin-75 (LOC724199)	0.00E+00	*Apis dorsata*	PREDICTED: extensin-like	2.00E-27
26	SSH-Forward-lib-36	320	*Nosema ceranae*	Polar tube protein 1 (PTP1) gene	2.00E-88	*Vibrio parahaemolyticus*	V-type ATPase subunit C	7.00E-14
27	SSH-Forward-lib-37	232	*Nosema ceranae*	Polar tube protein 2 (AAJ76_1900025375)	3.00E-85	*Nosema ceranae*	Polar tube protein 2	2.00E-34
28	SSH-Forward-lib-41	594	*Nosema ceranae*	Heat shock protein 90 (AAJ76_1170002375)	0.00E+00	*Nosema ceranae*	Heat shock protein 90	6.00E-122
29	SSH-Forward-lib-44	682	*Apis mellifera*	PREDICTED: *Apis mellifera* uncharacterized LOC100578731 (LOC100578731)	0.00E+00	*Apis mellifera*	PREDICTED: uncharacterized protein LOC100578731 isoform X1	6.00E-123
30	SSH-Forward-lib-46	411	*Nosema ceranae*	Hypothetical protein (AAJ76_500092411)	0.00E+00	*Nosema ceranae*	Actin	1.00E-87
31	SSH-Forward-lib-48	294	*Nosema ceranae*	Hypothetical protein (AAJ76_1900028347)	2.00E-129	*Nosema ceranae*	Hypothetical protein AAJ76_1900028347	5.00E-51
32	SSH-Forward-lib-49	85	*Apis dorsata*	PREDICTED: *Apis dorsata* V-type proton ATPase 16 kDa proteolipid subunit-like (LOC102679198)	3.00E-27	*Cyprinus carpio*	Hypothetical protein cypCar_00031997	6.00E-06
33	SSH-Forward-lib-51	231	*Nosema ceranae*	Hypothetical protein (AAJ76_2600030604)	7.00E-38	*Nosema ceranae*	Hypothetical protein AAJ76_2600030604	2.00E-05
34	SSH-Forward-lib-53	668	*Nosema ceranae*	Forkhead hnf3 transcription factor (AAJ76_500091146)	0.00E+00	*Nosema ceranae*	Forkhead hnf3 transcription factor	7.00E-126
35	SSH-Forward-lib-54	305	*Nosema ceranae*	Histone H4 (AAJ76_150002436)	8.00E-127	*Nosema ceranae*	Histone H4	1.00E-39
36	SSH-Forward-lib-55	731	*Apis mellifera*	PREDICTED: *Apis mellifera* facilitated trehalose transporter Tret1-like (LOC724874)	0.00E+00	*Apis mellifera*	PREDICTED: facilitated trehalose transporter Tret1-like isoform X4	2.00E-07
37	SSH-Forward-lib-56	279	*Apis mellifera*	*Apis mellifera* heat shock protein cognate 4 (Hsc70-4) (NM_001160050)	2.00E-117	*Bombus terrestris*	Heat shock protein cognate 70	6.00E-50
38	SSH-Forward-lib-57	420	*Apis mellifera*	*Apis mellifera* heat shock protein cognate 4 (Hsc70-4) (NM_001160050)	7.00E-164	*Melipona quadrifasciata*	Heat shock 70 kDa protein cognate 4	2.00E-69
39	SSH-Forward-lib-59	231	*Nosema ceranae*	Hypothetical protein (AAJ76_2000141845)	4.00E-93	*Nosema ceranae*	Hypothetical protein (AAJ76_2000141845)	6.00E-35
40	SSH-Forward-lib-60	668	*Nosema ceranae*	Forkhead hnf3 transcription factor (AAJ76_500091146)	0.00E+00	*Nosema ceranae*	Forkhead hnf3 transcription factor	1.00E-124
41	SSH-Forward-lib-61	811	*Nosema ceranae*	Hypothetical protein (AAJ76_500092411)	0.00E+00	*Nosema ceranae*	Actin	1.00E-84
42	SSH-Forward-lib-62	163	*Nosema ceranae*	Hypothetical protein (AAJ76_500092411)	1.00E-47	*Nosema ceranae*	Actin	6.00E-16
43	SSH-Forward-lib-63	651	*Nosema ceranae*	Hypothetical protein (AAJ76_500092411)	0.00E+00	*Nosema ceranae*	Actin	4.00E-141
44	SSH-Forward-lib-65	534	*Nosema ceranae*	Hypothetical protein (AAJ76_2000141845)	0.00E+00	*Nosema ceranae*	Hypothetical protein (AAJ76_2000141845)	3.00E-87
45	SSH-Forward-lib-66	615	*Nosema ceranae*	Hypothetical protein (AAJ76_500092411)	0.00E+00	*Nosema ceranae*	Actin	2.00E-133
46	SSH-Forward-lib-68	671	*Apis mellifera*	PREDICTED: *Apis mellifera* ankyrin repeat domain-containing protein 49-like (LOC408773)	0.00E+00	*Apis dorsata*	PREDICTED: ankyrin repeat domain-containing protein 49-like	3.00E-90
47	SSH-Forward-lib-70	163	*Nosema ceranae*	Hypothetical protein (AAJ76_500092411)	3.00E-49	*Nosema ceranae*	Actin	2.00E-16
48	SSH-Forward-lib-71	550	*Nosema ceranae*	Histone H3 (AAJ76_800037613)	0.00E+00	*Nosema ceranae*	Histone H3	4.00E-89
49	SSH-Forward-lib-73	248	*Apis mellifera*	PREDICTED: *Apis mellifera* ribosomal protein S7 (RpS7) (XM_624940) (LOC552564)	1.00E-101	*Bombus impatiens*	PREDICTED: 40S ribosomal protein S7	8.00E-22
50	SSH-Forward-lib-74	892	*Nosema ceranae*	14-3-3 protein 1-like protein (AAJ76_300017093)	0.00E+00	*Nosema ceranae*	14-3-3 protein 1-like protein	0.00E+00
51	SSH-Forward-lib-75	522	*Nosema ceranae*	Nucleolar transformer 2-like protein (AAJ76_1900029507)	0.00E+00	*Nosema ceranae*	Nucleolar transformer 2-like protein	2.00E-92
52	SSH-Forward-lib-76	521	*Apis mellifera*	PREDICTED: *Apis mellifera* uncharacterized (ncRNA) (LOC107964104)	0	*-*	-	-
53	SSH-Forward-lib-78	616	*Nosema ceranae*	Hypothetical protein (AAJ76_1900028347)	0.00E+00	*Nosema ceranae*	Hypothetical protein AAJ76_1900028347	7.00E-79
54	SSH-Forward-lib-79	293	*Nosema ceranae*	Hypothetical protein (AAJ76_1900028347)	5.00E-125	*Nosema ceranae*	Hypothetical protein AAJ76_1900028347	6.00E-51
55	SSH-Forward-lib-80	294	*Nosema ceranae*	Hypothetical protein (AAJ76_1900028347)	2.00E-124	*Nosema ceranae*	Hypothetical protein AAJ76_1900028347	6.00E-48
56	SSH-Forward-lib-82	226	*Nosema ceranae*	Adp-ribosylation factor (AAJ76_1600043340)	3.00E-100	*Nosema ceranae*	Adp-ribosylation factor	2.00E-40
57	SSH-Forward-lib-83	447	*Nosema ceranae*	DNA-directed RNA polymerase ii 16kda polypeptide (AAJ76_3400016364)	0.00E+00	*Nosema ceranae*	DNA-directed RNA polymerase ii 16kda polypeptide	6.00E-78
58	SSH-Forward-lib-84	227	*Nosema ceranae*	Histone H3 (AAJ76_800037613)	2.00E-90	*Nosema ceranae*	Histone H3	2.00E-38
59	SSH-Forward-lib-86	346	*Nosema ceranae*	Sec61beta (AAJ76_5600014344)	7.00E-129	*Nosema ceranae*	Sec61beta	2.00E-37
60	SSH-Forward-lib-87	354	*Nosema ceranae*	Heat shock protein 90 (AAJ76_1170002375)	2.00E-151	*Nosema ceranae*	Heat shock protein 90	2.00E-60
61	SSH-Forward-lib-88	886	*Nosema ceranae*	Elongation factor 1 (AAJ76_1200007387)	0.00E+00	*Nosema ceranae*	Elongation factor 1	0.00E+00
62	SSH-Forward-lib-90	252	*Nosema ceranae*	Ubiquitin (AAJ76_1300017036)	3.00E-90	*Nosema ceranae*	Ubiquitin	3.00E-38
63	SSH-Forward-lib-94	529	*Nosema ceranae*	60s ribosomal protein L23 (AAJ76_200068052)	0.00E+00	*Nosema ceranae*	60S ribosomal protein L23	1.00E-101
64	SSH-Forward-lib-95	320	*Hottentotta judaicus*	Hottentotta judaicus clone Hj0016 gamma(m)-buthitoxin-Hj1c pseudogene mRNA (HQ288097)	1.00E-14	*Hottentotta judaicus*	Gamma-buthitoxin-Hj1a	3.00E-18
65	SSH-Forward-lib-96	401	*Nosema ceranae*	Histone H4 (AAJ76_150002436)	1.00E-151	*Nosema ceranae*	Histone h4	5.00E-52

**Table 3 insects-11-00199-t003:** Analysis of 53 differential expressed sequence tags (ESTs) identified from reverse cDNA library.

No	EST Number	Insert Sequence Length (bp)	NCBI_BLAST_N			NCBI_BLAST_X		
			Species	Gene Name	E-Value	Species	Gene Name	E-Value
1	SSH-Reverse-lib-1	387	*Colletes stepheni*	*Colletes stepheni* isolate CSTM4 internal transcribed spacer 1, partial sequence (GU132310)	3.00E-03	*Elephantulus edwardii*	Histone deacetylase complex subunit SAP130-like	1.5
2	SSH-Reverse-lib-2	216	*Apis mellifera*	PREDICTED: *Apis mellifera* facilitated trehalose transporter Tret1-like (LOC724874)	3.00E-50	*Apis mellifera*	Facilitated trehalose transporter Tret1 isoform X1	10.00E-15
3	SSH-Reverse-lib-3	156	*Apis mellifera*	PREDICTED: *Apis mellifera* chitin synthase chs-2 (LOC412215)	6.00E-51	*-*	-	-
4	SSH-Reverse-lib-4	467	*Apis mellifera*	PREDICTED: *Apis mellifera* histone H2A-like (LOC552322)	0.00E+00	*Nasonia vitripennis*	PREDICTED: histone H2A	4.00E-57
5	SSH-Reverse-lib-5	829	*Apis mellifera*	PREDICTED: *Apis mellifera* early nodulin-75 (LOC724199)	0.00E+00	*Apis dorsata*	PREDICTED: extensin-like	2.00E-28
6	SSH-Reverse-lib-6	331	*Bombus impatiens*	PREDICTED: *Bombus impatiens* zinc finger protein 665-like (LOC100740332)	1.00E-39	*-*	-	-
7	SSH-Reverse-lib-7	488	*Apis mellifera*	PREDICTED: *Apis mellifera* protein SMG9 (LOC411774)	0.00E+00	*Apis mellifera*	PREDICTED: protein SMG9-like	3.00E-95
8	SSH-Reverse-lib-10	382	*Apis mellifera syriaca*	*Apis mellifera* syriaca mitochondrion ( KP163643)	5.00E-38	*-*	-	-
9	SSH-Reverse-lib-12	233	*Megachile rotundata*	PREDICTED: *Megachile rotundata* probable maleylacetoacetate isomerase 2 (LOC100882531)	2.00E-14	*-*	-	-
10	SSH-Reverse-lib-14	192	*Apis mellifera*	PREDICTED: *Apis mellifera* caspase-like (LOC411381)	3.00E-69	*Apis mellifera*	PREDICTED: caspase-like	7.00E-29
11	SSH-Reverse-lib-15	546	*Apis dorsata*	PREDICTED: *Apis dorsata* nuclease-sensitive element-binding protein 1-like (LOC102681001)	0.00E+00	*-*	-	-
12	SSH-Reverse-lib-16	218	*Ceratitis capitata*	*Ceratitis capitata* clone 17a mRNA ( DQ406807)	0.019	*-*	-	-
13	SSH-Reverse-lib-17	629	*Apis florea*	PREDICTED: *Apis florea* histone H3.3 (LOC100869036)	4.00E-30	*-*	-	-
14	SSH-Reverse-lib-18	166	*Hottentotta judaicus*	*Hottentotta judaicus* clone Hj0016 gamma(m)-buthitoxin-Hj1c pseudogene mRNA (HQ288097)	2.00E-12	*-*	-	-
15	SSH-Reverse-lib-19	246	*Apis mellifera*	PREDICTED: *Apis mellifera* probable alpha-ketoglutarate-dependent dioxygenase ABH4-like (LOC551104)	6.00E-98	*Apis mellifera*	PREDICTED: alpha-Ketoglutarate-dependent dioxygenase alkB homolog 4-like	1.00E-25
16	SSH-Reverse-lib-20	658	*Apis mellifera*	PREDICTED: *Apis mellifera* facilitated trehalose transporter Tret1-like (LOC724874)	0.00E+00	*Apis mellifera*	PREDICTED: facilitated Trehalose transporter Tret1-like isoform X1	7.00E-81
17	SSH-Reverse-lib-21	976	*Boechera divaricarpa*	*Boechera divaricarpa* GSS (HF949778)	0.00E+00	*Apis mellifera*	PREDICTED: transcription Initiation factor IIA subunit 1 isoform 1	8.00E-48
18	SSH-Reverse-lib-22	469	*Apis mellifera*	PREDICTED: *Apis mellifera* ribosomal protein L3 (RpL3) (LOC552445)	0.00E+00	*Apis mellifera*	PREDICTED: 60S ribosomal protein L3	1.00E-80
19	SSH-Reverse-lib-25	482	*Apis mellifera*	PREDICTED: *Apis mellifera* UNC93-like protein MFSD11-like (LOC552407)	0.00E+00	*Apis mellifera*	PREDICTED: UNC93-like protein MFSD11-like	8.00E-77
20	SSH-Reverse-lib-26	273	*Apis mellifera*	PREDICTED: *Apis mellifera* vegetative cell wall protein gp1 (LOC726855)	1.00E-108	*Apis mellifera*	PREDICTED: vegetative cell wall protein gp1-like isoform X1	8.00E-30
21	SSH-Reverse-lib-28	440	*Apis mellifera*	PREDICTED: *Apis mellifera* uncharacterized (LOC102653963)	0	*-*	-	-
22	SSH-Reverse-lib-32	556	*Apis mellifera*	PREDICTED: *Apis mellifera* NPC intracellular cholesterol transporter 2 (LOC724386)	0.00E+00	*Apis mellifera*	PREDICTED: protein NPC2 homolog	1.00E-78
23	SSH-Reverse-lib-35	269	*Apis mellifera*	PREDICTED: *Apis mellifera* vegetative cell wall protein gp1 (LOC726855)	1.00E-100	*Apis mellifera*	PREDICTED: vegetative cell wall protein gp1-like isoform X1	3.00E-25
24	SSH-Reverse-lib-38	318	*Apis mellifera*	PREDICTED: *Apis mellifera* RNA-dependent helicase p72 (LOC411250)	6.00E-131	*-*	-	-
25	SSH-Reverse-lib-39	233	*Megachile rotundata*	PREDICTED: *Megachile rotundata* probable maleylacetoacetate isomerase 2 (LOC100882531)	2.00E-14	-	-	-
26	SSH-Reverse-lib-41	454	*Apis mellifera*	PREDICTED: Apis mellifera chitotriosidase-1-like (LOC100577156)	3.00E-30	*Apis mellifera*	PREDICTED: chitotriosidase-1-like isoform X3	2.00E-10
27	SSH-Reverse-lib-45	377	*Colletes stepheni*	*Colletes stepheni* isolate CSTM4 internal transcribed spacer 1 (GU132310 )	2.00E-04	*Elephantulus edwardii*	PREDICTED: histone Deacetylase complex subunit SAP130-like	0.99
28	SSH-Reverse-lib-49	386	*Colletes stepheni*	*Colletes stepheni* isolate CSTM4 internal transcribed spacer 1 (GU132310 )	2.00E-04	*Elephantulus edwardii*	PREDICTED: WD repeat-containing protein 27	4.4
29	SSH-Reverse-lib-52	247	*Apis mellifera*	PREDICTED: Apis mellifera RNA-dependent helicase p72 (LOC411250)	1.00E-101	*Apis dorsata*	DEAD-box ATP-dependent RNA helicase 20-like	0
30	SSH-Reverse-lib-58	213	*Apis mellifera*	PREDICTED: *Apis mellifera* phosphotidylinositol 3 kinase 21B ortholog (LOC408577)	4.00E-80	*Curtobacterium sp.*	Hypothetical protein	4.8
31	SSH-Reverse-lib-59	327	*Apis mellifera*	PREDICTED: *Apis mellifera* ribosomal protein S7 (RpS7) (LOC552564)	2.00E-138	*Apis mellifera*	PREDICTED: 40S ribosomal protein S7	2.00E-58
32	SSH-Reverse-lib-60	199	*Apis mellifera*	PREDICTED: *Apis mellifera* solute carrier organic anion transporter family member 5A1-like (LOC409650)	1.00E-60	*-*	-	-
33	SSH-Reverse-lib-61	237	*Apis mellifera*	PREDICTED: *Apis mellifera* Jun-related antigen (Jra) (LOC726289)	7.00E-91	*Salmonella enterica*	Hypothetical protein, partial	2.4
34	SSH-Reverse-lib-63	274	*Apis mellifera*	PREDICTED: *Apis mellifera* vegetative cell wall protein gp1 (LOC726855)	1.00E-107	*Apis mellifera*	PREDICTED: vegetative cell wall protein gp1-like isoform X1	3.00E-18
35	SSH-Reverse-lib-67	377	*Colletes stepheni*	*Colletes stepheni* isolate CSTM4 internal transcribed spacer 1 (GU132310 )	0.003	*Danio rerio*	glutamate receptor 3 precursor	5.3
36	SSH-Reverse-lib-68	386	*Colletes stepheni*	*Colletes stepheni* isolate CSTM4 internal transcribed spacer 1 (GU132310 )	2.00E-04	*-*	-	-
37	SSH-Reverse-lib-69	376	*Colletes stepheni*	*Colletes stepheni* isolate CSTM4 internal transcribed spacer 1 (GU132310 )	0.003	*Nothobranchius furzeri*	PREDICTED: histone deacetylase complex subunit SAP130	4.7
38	SSH-Reverse-lib-70	275	*Apis mellifera*	PREDICTED: *Apis mellifera* upstream activation factor subunit spp27-like (LOC408520)	3.00E-109	*-*	-	-
39	SSH-Reverse-lib-71	377	*Colletes stepheni*	*Colletes stepheni* isolate CSTM4 internal transcribed spacer 1 (GU132310 )	2.00E-04	*Elephantulus edwardii*	PREDICTED: WD repeat-containing protein 27	4.9
40	SSH-Reverse-lib-72	329	*Apis mellifera*	PREDICTED: *Apis mellifera* transcription factor mblk-1-like (Mblk-1)	1.00E-139	*-*	-	-
41	SSH-Reverse-lib-74	563	*Apis dorsata*	PREDICTED: *Apis dorsata* uncharacterized (LOC102671476)	0.00E+00	*Apis dorsata*	PREDICTED: uncharacterized protein LOC102671476	6.00E-95
42	SSH-Reverse-lib-75	281	*Apis mellifera*	Phosphatidylinositol 4-phosphate 5-kinase type-1 alpha (LOC724991)	4.00E-115	*-*	-	-
43	SSH-Reverse-lib-79	946	*Apis mellifera*	PREDICTED: *Apis mellifera* protein FAM76A-like (LOC408294)	0.00E+00	*Apis mellifera*	PREDICTED: ATPase family AAA domain-containing protein 2-like	1.00E-63
44	SSH-Reverse-lib-84	383	*Apis mellifera*	PREDICTED: *Apis mellifera* ribosomal protein L21 (RpL21) (LOC726056)	3.00E-134	*Apis mellifera*	PREDICTED: 60S ribosomal protein L21	3.00E-47
45	SSH-Reverse-lib-87	404	*Apis mellifera*	PREDICTED: *Apis mellifera* protein split ends (LOC412243)	3.00E-167	*-*	-	-
46	SSH-Reverse-lib-92	532	*Hottentotta judaicus*	Gamma(m)-buthitoxin-Hj1c pseudogene mRNA (HQ288097)	2.00E-14	*Apis cerana*	Hypothetical protein APICC_07509	5.6
47	SSH-Reverse-lib-94	487	*Colletes stepheni*	*Colletes stepheni* isolate CSTM4 internal transcribed spacer 1 (GU132310 )	3.00E-04	*Nothobranchius furzeri*	PREDICTED: histone deacetylase complex subunit SAP130	4.2

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
