# Peer review of "Screening of Differentially Expressed Microsporidia Genes from Nosema ceranae Infected Honey Bees by Suppression Subtractive Hybridization"

_insects, 2020, doi:10.3390/insects11030199_

Round 1

Reviewer 1 Report

The manuscript "Screening of differentially expressed microsporidia genes from Nosema ceranae infected honey bee by suppression subtractive hybridisation" by Chang et al shows gene expression from N. ceranae during infection of honey bees. This is interesting as most previous studies have focused exclusively on the bees, themselves. The introduction is thorough and clear, the methods are sufficient and the results are well presented. However, the manuscript is marred by several language issues. After a thorough language editing, I believe it can be submitted as it stands (re. figures, results, etc).

Author Response

The manuscript "Screening of differentially expressed microsporidia genes from Nosema ceranae infected honey bee by suppression subtractive hybridisation" by Chang et al shows gene expression from N. ceranae during infection of honey bees. This is interesting as most previous studies have focused exclusively on the bees, themselves. The introduction is thorough and clear, the methods are sufficient and the results are well presented. However, the manuscript is marred by several language issues. After a thorough language editing, I believe it can be submitted as it stands (re. figures, results, etc).

Ans: Thank you for your comments, it really encourages this work. We have also checked throughout this manuscript and improved some language issues as well.

Reviewer 2 Report

The honeybee (Apis mellifera) is an essential pollinator worldwide. However, it’s usually infested by different pathogens, including parasites, mites, microsporidia, bacteria, and viruses, which result in bee losses, and may also be associated with bee disappearing as called the colony collapse disorder. In this study, the author identified the differentially expressed genes (DEGs) of the honeybee in response to the infection of N. ceranae using the suppression subtractive hybridization (SSH) method. The results could better interpret how honeybee response to N. ceranae molecularly. 

  1. The authors stated that a high percentage of DEGs involved in catalytic activity and metabolic processes of the honeybee in response to the infection of N. ceranae. However, the infection of other pathogens could usually induce immune response genes. I am concerned if the authors lost too many genes after SSH process. Do the authors identify highly up- or down-regulated genes that are associated with honeybee immunity in response to the infection of N. ceranae.
  2. The statement of “highly up-regulated N. ceranae specific genes” is not correct. Up-regulated genes are the honeybee responsive genes that are up-regulated in response to the infection of N. ceranae. Since these genes are all related to N. ceranae genes, I would use “highly expressed” but not “highly up-regulated” in the manuscript. 
  3. Do bacteria contaminate the ESTs 25, 32, and 53 on Page 15-19 in Table 3?

Some minor changes:

Page 1, lines 39-47: is this number ($14 billion to agriculture; Reductions of honeybee populations were estimated at 23% over the winter of 2006-2007 and at 36% over the winter of 2007-2008) globally, or in the USA, or Taiwan.

Page 4, line 166: the E-value for the “best hit” match.

Page 4, lines 179-180: delete at 42oC for 1.5 h.

Page 4, line 182: Please insect the protocol and the conditions for RT-qPCR.

Page 5, line 191: Change Figure 1B into Figure 1D.

Page line 198: Change Figure 1D into Figure 1C. Which’s the X and Y axle in Figure 1C?

Page 6 Figure 2A&B: correct (bp                                                     

                                                       ) into (bp).

Page 6, Table 1: This is not the statistics. Please correct the title.

Page 6-7, lines 241-245: Do authors find the immune response and oxidative genes?

Page 9: Please combine Figures 5 A and 5 B in one by using two segments (….//….)for the X-axis.

Author Response

Comments and Suggestions for Authors

The honeybee (Apis mellifera) is an essential pollinator worldwide. However, it’s usually infested by different pathogens, including parasites, mites, microsporidia, bacteria, and viruses, which result in bee losses, and may also be associated with bee disappearing as called the colony collapse disorder. In this study, the author identified the differentially expressed genes (DEGs) of the honeybee in response to the infection of N. ceranae using the suppression subtractive hybridization (SSH) method. The results could better interpret how honeybee response to N. ceranae molecularly. 

  1. The authors stated that a high percentage of DEGs involved in catalytic activity and metabolic processes of the honeybee in response to the infection of N. ceranae. However, the infection of other pathogens could usually induce immune response genes. I am concerned if the authors lost too many genes after SSH process. Do the authors identify highly up- or down-regulated genes that are associated with honeybee immunity in response to the infection of N. ceranae.

Ans: Thank you for your comment. As the matter of fact that the sequencing information, which derived from SSH process might be influenced by the number of sequenced ESTs and the SSH process (i.e. hybridization), thus the sequences from SSH library in this study would be the highly differentially expressed ESTs. It is good comments, which we will extend the research by deep sequencing or NGS to further investigate. Based on the sequenced ESTs, there were few host immune-related genes up-regulated during N. ceranae infection, while the immune-related genes (Jun-related antigen (Jra), pi3k, etc.) of host were identified from the revere subtractive EST library. Therefore, the suppression of host immune-related gene was found after N. ceranae infection.

We have added the description as below “There were only few host genes showed the dramatically responses of up-regulated during N. ceranae infection, while most of the up-regulated host genes were identified from the rscl, suggested the suppression of host gene expression by N. ceranae infection. Based on the result, several immune-related genes (i.e., Jra, pi3k) of honeybees were identified from the rscl.”

  1. The statement of “highly up-regulated N. ceranaespecific genes” is not correct. Up-regulated genes are the honeybee responsive genes that are up-regulated in response to the infection of N. ceranae. Since these genes are all related to N. ceranaegenes, I would use “highly expressed” but not “highly up-regulated” in the manuscript. 

Ans: Thank you for this great comment. We have checked the relative parts in this manuscript and corrected “highly up-regulated” N. ceranae specific genes to “highly expressed” N. ceranae specific genes. The corrections were highlighted by “Track Changes" function in the manuscript.

  1. Do bacteria contaminate the ESTs 25, 32, and 53 on Page 15-19 in Table 3?

 Ans: We have checked these sequences and also rechecked the sequences in Table 2 and 3. After hybridization, the ESTs were subjected into the bacterial cloning system for sequencing. During the EST library construction, the insert DNA fragments less than 100bp or without inserts or any significant similarity should be removed, therefore, the ESTs 19, 25, 32, 48, 49 and 53 were removed. We have also corrected the relative parts throughout this manuscript.     

Some minor changes:

Page 1, lines 39-47: is this number ($14 billion to agriculture; Reductions of honeybee populations were estimated at 23% over the winter of 2006-2007 and at 36% over the winter of 2007-2008) globally, or in the USA, or Taiwan.

Ans: This data was estimated in North America. We have corrected as “…to agriculture annually in North America.”

Page 4, line 166: the E-value for the “best hit” match.

Ans: We have corrected it as “Each unique sequence was tentatively assigned GO classification based on annotation of the E-value for the “best hit” match. Finished. Thanks for reminding.”

Page 4, lines 179-180: delete at 42oC for 1.5 h.

Ans: Thank you for your comment. We have deleted it.

Page 4, line 182: Please insect the protocol and the conditions for RT-qPCR.

Ans: We have inserted the protocol for RT-qPCR as following, “…The qRT-PCR was performed as follows: 95°C for 3 min, 40 cycle at 95°C for 10s, 59°C for 30s, 95°C for 10s.”

Page 5, line 191: Change Figure 1B into Figure 1D.

Ans: We have corrected it.

Page line 198: Change Figure 1D into Figure 1C. Which’s the X and Y axle in Figure 1C?

Ans: We have corrected it. The X axis is the survival time (Day) and Y-axis is the survival probability. We have also added in Figure 1C as below.

Page 6 Figure 2A&B: correct (bp) into (bp).

Ans: Thank you for this comment, we have corrected it.

Page 6, Table 1: This is not the statistics. Please correct the title.

Ans: We have changed the title to Table 1. Information for the expression sequence tags (ESTs).

Page 6-7, lines 241-245: Do authors find the immune response and oxidative genes?

Ans: Based on the sequenced ESTs, there were few host immune-related genes up-regulated during N. ceranae infection, while the immune-related genes (Jun-related antigen (Jra), pi3k, etc.) of host were identified from the revere subtractive EST library. Therefore, the suppression of host immune-related gene was found after N. ceranae infection. We have added the description as below “There were only few host genes showed the dramatically responses of up-regulated during N. ceranae infection, while most of the up-regulated host genes were identified from the rscl, suggested the suppression of host gene expression by N. ceranae infection. Based on the result, several immune-related genes (i.e., Jra, pi3k) of honeybees were identified from the rscl.”

Page 9: Please combine Figures 5 A and 5 B in one by using two segments (….//….)for the X-axis.

Ans: Thank you for this comment. We have combined the Figures 5 A and 5 B as below, and it looks much better than before. Besides, we have also corrected the related citations of this figure.
